# Security Establishment in ADS-B by Format-Preserving Encryption and Blockchain Schemes

Jamal Habibi Markani [1,*], Abdessamad Amrhar [1], Jean-Marc Gagné [1] and René Jr Landry [2]

[1]  LASSENA Laboratory, École de Technologies Supérieure (ÉTS), Montréal, QC H3C 1K3, Canada
[2]  Department of Electrical Engineering, École de Technologie Supérieure, Montréal, QC H3C 1K3, Canada
*  Correspondence: jamal.habibi@lassena.etsmtl.ca

**Abstract:** In the next generation modernization plan, the automatic dependent surveillance-broadcast (ADS-B) system plays a pivotal role. However, the ADS-B's low level of security and its vulnerabilities have raised valid concerns. The main objectives of this paper are to highlight the limitations of legacy ADS-B systems and to assess the feasibility of using Format-preserving (F), Feistel-based encryption (F), with multiple implementation variances (X) (FFX) algorithms, for enhancing ADS-B's security. The offered solution is implemented in a standard software-defined radio (SDR) ADS-B to be utilized in real-time applications. Furthermore, a new proposed blockchain scheme is used as a secured database to manage the cipher key. The metric of message entropy is used to assess an algorithm's ability to confuse and diffuse predictable ADS-B messages; correlation and serial correlation of plain data and cipher data are deployed to evaluate the proposed method's security level. The authors provide both MATLAB simulations and flight test outcomes to demonstrate the feasibility of this approach. Based on our security analysis, ADS-B information can be kept confidential through our scheme. The performance evaluation results reveal that the proposed scheme is achievable, compatible, and efficient for the avionics industry.

**Keywords:** ADS-B; NextGen; SDR; format-preserving encryption; FFX; blockchain; BladeRF

## 1. Introduction

The ADS-B system broadcasts an aircraft's identification, position, velocity, and intent to other aircraft and ground stations. However, the introduction of these new technologies in the aviation industry has introduced new network weaknesses linked to their internet connection and the compatibility between systems [1].

Certain technological advancements, particularly the introduction of Software-Defined Radio (SDR) systems, enable attackers to utilize low-cost RF transmission and receiving. ADS-B transmits data in an open, unencrypted format to allow other aircraft to be informed of the sender's location. On the other hand, accessibility had been a safety requirement. Therefore, it lacks protection features, rendering it susceptible to a variety of assaults [2]. Researchers have shown that existing hardware and software can easily undermine the security of ADS-B systems [3]. Table 1 summarizes ADS-B's possible attack types, methods, and severity levels, and the requirements they impact most severely [4]. Therefore, the conventional ADS-B's data confidentiality, authenticity and integrity can be impacted by possible attacks. This problem might lead to severe consequences [4]. Although a number of solutions are offered to overcome the issue, either they were led to increase the ADS-B message length and reduce ADS-B air-traffic capacity, or the proposals were not implemented and examined in operational environments. Furthermore, some of the solutions impacted ADS-B performance, such as the degradation of ADS-B sensitivity. Additionally, cipher key exchange was not properly addressed. The proposed solutions are an imposed extra load to ADS-B messages by adding an authentication procedure, peer-to-peer communication, or a key data payload. These approaches lead to the reduction of ADS-B capacity and increased the system complexity and load.

**Table 1.** ADS-B attack classification.

| Type of Attack | Attack Methods | Attack Severity Level | Impacted Requirement |
|---|---|---|---|
| Passive attacks | Eavesdropping | Low | Confidentiality |
| | Jamming | Medium | Availability |
| Active attacks | Message Injection | High | Authentication |
| | Message Deletion | High | Integrity |
| | Message Modification | High | Integrity |

Very recent research was undertaken to deal with ADS-B security concerns, as referenced below:

- Blockchain based ADS-B: The proposed model's objective was to deploy a blockchain model and benefit from its advantages to secure sensitive data [5–8];
- Machine learning model: Various types of neural networks were offered to tackle message injection attacks [9,10]; and
- Holistic framework: The solution used a cryptographic module based on a keyed-Hash Message Authentication Code (HMAC) to protect data integrity and authenticity [11];

Detailed information of each method is provided in the literature review section.

This research covers the design and implementation of a secured ADS-B (SADS-B) system using a software-defined radio (SDR) architecture. It explores the hardware architecture and configuration, presents a format-preserving encryption (FPE) scheme, and shows how blockchain technology can be used for leveraging ADS-B security and how BladeRF functions as a transceiver. The results achieved using this secured system are compatibility, achievability, and high security. Given the current, un-secured ADS-B systems, this work is motivated by the following issues:

- Most of the recommended solutions are still in the experimental stage, being tested in controlled settings, and have yet to be implemented [4];
- Deploying FPE algorithms offers certified encryption for the protection of ADS-B messages without any changes in the message's length;
- While the use of FPE algorithms has been proposed as a security solution to benefit from its advantages [12], it has not yet been implemented and has not been challenged in operational environments such as implementation in SDR and experimentation in flight tests; and
- A permissioned blockchain system helps the operational environment not only to exchange the encryption key among nodes (ADS-B systems), but also to store important information.

In this study, FPE algorithms' advantages are described and then implemented in both MATLAB Simulink and C++. Their evaluation was assessed under real flight data to obtain a comprehensive observation of the algorithm as well as of the ADS-B performance. A private permissioned blockchain was used to distribute keys among SADS-B systems and to store sensitive positioning data. The system's computational load and performance are compared with a certified ADS-B technology. The results of these experiments show that the proposed scheme offers high-level security and compatibility, and reveal that it is easy to implement.

## 2. Literature Review

ADS-B security issues have been a hot research topic for at least the past 10 years. Several encryption techniques have been proposed for the protection of ADS-B data [4]. These encryption schemes are classified as symmetric or asymmetric encryption based on the similarity and dissimilarity of the encryption and the decryption keys.

A digital signature is a mathematical scheme for verifying the authenticity and the inverse function of public-key cryptography. The data is encrypted with a private key by the sender, and can then be decrypted with a public key by the receiver. A lightweight identity-based batch verification (IBV) signature scheme is suggested by Zhou et al. [13] to establish

security via batch message identification. This IBV system has considerable resistance to replay attacks. According to their results, the method has a low computational load and a low transmission overhead. However, Zhou et al.'s proposal was not put into practice. In a broader work, Wesson et al. [14] examined the merits and drawbacks of both symmetric and asymmetric encryptions, digital signatures, and related key management. Their study concluded that symmetric encryption key management is difficult to control, and that the only feasible encryption technique is asymmetric encryption. Specifically, they determined that the Elliptic Curve Digital Signature Algorithm (ECDSA) signature length is the smallest length that should be applied. The ECDSA signature length is 448 bits, which makes the overall message length 560 bits. In the proposed approach, the message is divided into nine segments; the first segment has 112 bits, followed by eight 56-bit segments. Next, the signature is offered to be broadcast, either by other flight radio channels to support ADS-B, or via the distance measuring equipment (DME) channel that can be utilized to transmit ADS-B signed data. Both approaches reduced the delay of the transmitted message, but the message's length was still too long. These approaches are not recommended.

Contrary to symmetric encryption, peer-to-peer communication is not needed for the key exchange in the public-key encryption approach. A message authentication procedure built on elliptic curve cryptography (ECC) and the X.509 certificate was proposed by Strohmeier et al. [15] and Ziliang et al. [16]. Their proposal exploits the ECSDA technique to produce signatures for ADS-B messages. This approach offers ADS-B data integrity and non-repudiation. Although this solution is not costly, it is less scalable than the existing ADS-B system. Baek [17] introduced a phased hybrid encryption method to fix the symmetric encryption key exchange limitation and to enhance encryption performance. This method applies encryption in two stages: key encryption and data encryption. Key encryption utilizes identity-based encryption, and therefore certificate management is not required. Data encryption employs symmetric encryption to optimize its advantages. Data encryption can thus be performed quicker and more effectively than public-key encryption. In practice, the sender needs to have prior knowledge of the ground stations or other airplanes, and data can be sent to only one receiver at a time, which is not appropriate.

The Message Authentication Code (MAC) encryption technique was proposed by Samuelson et al. [18] to protect ADS-B messages. In MAC, all of the nodes perform authentication by appending the authentication code to the plain ADS-B data to produce an authentication identity for every node; irrespective of the verification result, all nodes can receive the data. Therefore, this approach maintains the openness feature of the ADS-B system. Taking into consideration that the ADS-B ciphertext with public key encryption has a 1024-bit length, and Universal Access Transceiver (UAT) data has only a 272-bit length, Samuelson et al. [18] conclude that encryption of the data by applying the public key is a viable alternative. After investigating the possibility of encrypting the ADS-B data, Jochum et al. [19] also suggested the same encryption scheme, but both proposals only provided a high-level encryption structure; no detailed encryption schemes were offered.

In 2018, LASSENA's researchers presented an approach to secured ADS-B by utilizing the combination of phase shift keying (PSK) modulation with the Pulse Position Modulation (PPM) of legacy ADS-B [20]. The aim was to increase the payload efficiency of the standard ADS-B. PSK modulation provides extra payloads to address the digital signature data requirement. Although the system keeps the openness feature of standard ADS-B, this scheme's PSK modulation is less robust than PPM modulation, and so the system may be more vulnerable under high-traffic.

In the article [9], machine learning algorithms such as Support Vector Machine (SVM) are proposed in order to detect three types of message injection attack: path modification, ghost aircraft injection, and velocity drift. The OpenSky network dataset is used to train and validate the proposed model performance [21]. According to the results, the model accuracy for detecting attacks is over 95%. Another new study is that referred to in [22], which had the objective of protecting against message injection attacks. The technique is based on physical layer signal analysis, which involves estimating the range and direction

of the signal source. A multi-channel receiver on ADS-B is part of the proposed system architecture, along with a directional slotted antenna system. The results are presented as the probability of detection versus the signal's direction of arrival (DoA).

In [10], a supervised deep learning approach was used to identify injection attacks that changed the ADS-B messages' positioning and velocity elements. The OpenSky dataset is used to train and evaluate the Long Short-Term model for prediction of any error in flight path data. However, a simulation result shows about 99 percent of accuracy; this approach can be used to detect message injection threats.

Recently, a holistic framework was proposed to overcome the security concerns of ADS-B [11]. The solution used a cryptographic module based on a keyed-Hash Message Authentication Code (HMAC) to protect data integrity and authenticity. In addition, the cryptographic keys exchange mechanism is defined to share the key to the ADS-B nodes. However, the proposed model tries to address security issues, and it imposes an extra load on the ADS-B traffic, which causes capacity degradation.

To address the limited payload capacity (112 bits) of the legacy ADS-B, and because ciphering messages that use a conventional encryption algorithm lead to longer cipher texts, Finke et al. [12] proposed a format-preserving encryption (FPE) method to encrypt ADS-B data and to reduce ciphertext length. Fixed-length messages and confidential fixed-length data such as credit card numbers and social insurance numbers do not fit into the standard block size (64 and 128 bits), and so the FPE method is primarily deployed to encrypt data without changing the data format [23]. Contrary to traditional packet ciphering, the reserved format encryption does not expand the data. The ciphertext keeps the format encryption in the same format as the plaintext, and thus this method minimizes encryption impact on the ciphertext length. Unlike the Finke et al. [12] method that encrypts all of the ADS-B data, Yang et al. [24] exploit the FPE scheme to encrypt only the ICAO24 Aircraft Address (AA) field of the ADS-B message. Thus, this method not only preserves the openness of the ADS-B system, but it also secures the aircraft's identity data against unauthorized users. It also protects aircraft against reconnaissance attacks. This approach provides confidentiality to some extent, as not all data is encrypted. Agbeyibor [25] conducted a survey on the applicability of several types of FPE algorithms to apply ADS-B message encryption and examined their security and performance. Our research investigates the FPE method's security and performance based on real flight data, and it proposes a solution for how to apply the FPE scheme in a software-defined radio platform.

There has been a strong trend recently among researchers and developers to use blockchain to increase avionics security. Blockchain has been exploited in several ways to address ADS-B security requirements. ATMChain is a blockchain-based framework presented by Xin Lu and Zhijun Wu [5] for the security of cyber-physical systems (CPS) in the Air Traffic Management (ATM) system. Their proposed model is based on an ATM-focused approach, in which the distributed blockchain network is used to construct the ATM-CPS security framework. ADS-B sensors compose one of the different modules, accompanied by satellites, radar units, and other devices. Farah Hasin et al. suggest using the Hyperledger Fabric platform to deal with the security risks related to data link transmissions over ADS-B [8]. Their proposed ATM system consists of participants, such as various types of aircraft, which are divided into Zones A and B and are connected by a consensus blockchain network. However, the study is limited to the theoretical aspect.

Another approach utilizing blockchain was investigated by Arushi Arora et al. [6]. They propose a solution according to the aircraft's registration and their connection to the ground station. Their algorithm consists of three steps: first, a registered aircraft is considered as one transaction in the ledger that is authorized by ATC, then the aircraft is authenticated when the first GS sees it, and finally, control is transferred from one GS to another to perform the handover.

Reisman et al. describe a blockchain-based public key infrastructure (PKI) solution to address ADS-B vulnerabilities [7]. Their proposed aviation blockchain infrastructure (ABI) serves as a link between technologies that are used in the air and those used on the ground.

However, the limitations of real-time processing and the management of high numbers of transactions is not highlighted in PKI-based blockchain.

These papers reveal the possibilities for using blockchain technology in ADS-B systems and provide direction and guidelines for our work. Although blockchain-related articles have proposed high-level security architectures for the entire ATM system or among nodes, they have not provided detailed information on how to implement it in operational environments.

## 3. Methodology and Solutions

This section describes the design and implementation of SADS-B and offers a solution for key exchange. Format-preserving (F), Feistel-based encryption (F) with multiple implementation variances (X) is referred to as FFX [23]. The FFX algorithm, SADS-B implementation, and blockchain-based key exchange solution are described below.

### 3.1. FFX Algorithm

The Feistel structure serves as the foundation for all FFX algorithms, including FF1 and FF3-1, which have been established as the US National Institute of Standards and Technology (NIST) standard.

A Feistel structure is created when reversible transformation is repeated numerous times, known as "rounds". The transformation is divided into three phases. First, data is divided into two segments, and a keyed function named the round function is applied to one segment of the data. For the next round, the roles of the two parts are switched to alter another segment of the data. The function F generates a repeatable hash-like value by using a user-specified symmetric block cipher. Figure 1 below shows the framework for both encryption and decryption for ten rounds, although FF1 and FF3-1 have ten and eight rounds, respectively.

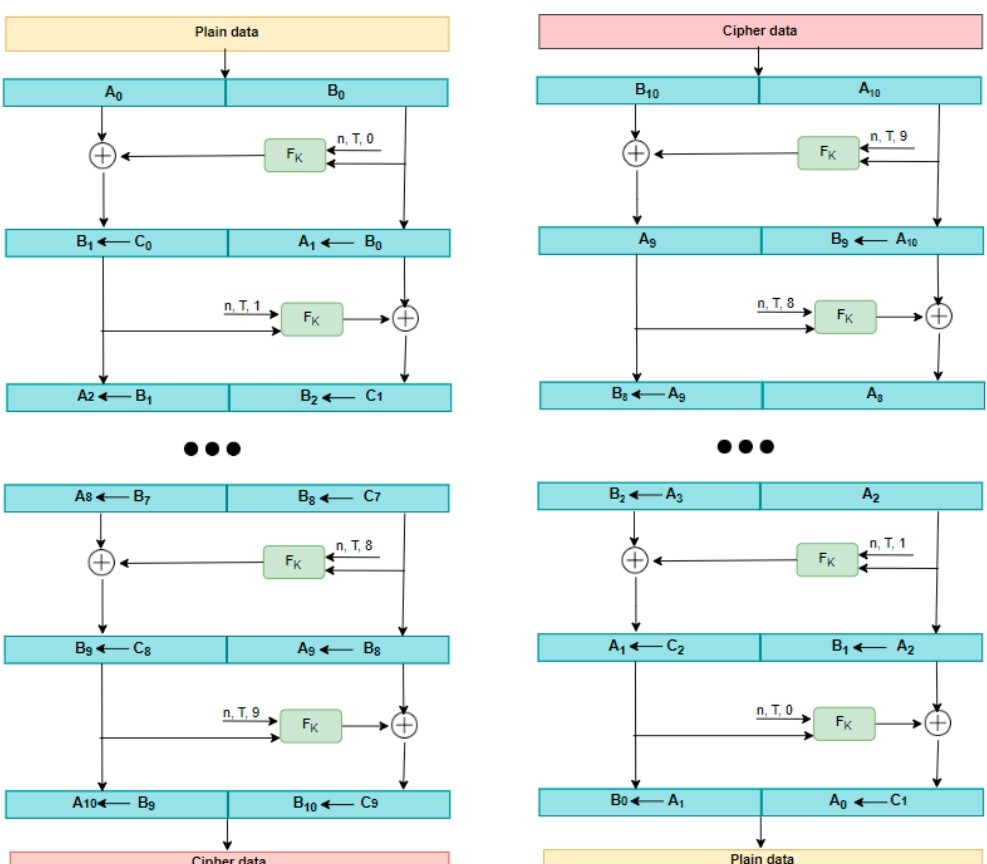

**Figure 1.** Encryption (**left**) and decryption (**right**) structure of FF1 and FF3-1 [23].

Two strings of characters serve as the input (and output) data for each round; for FF1 and FF3-1, these strings will be numerals. The two strings are equal in length, and n represents the overall number of characters. The round function, denoted by FK in Round i, is applied to one of the input strings, marked by Bi, with the extra inputs of length n, tweak T, and round number i. (In Figure 1, these three extra inputs (n, T, and i are shown as the super-posed arrow with the proper values for i.) By applying modular addition—indicated by the plus sign ⊕—on the numbers the strings stand for, the result is utilized to change the other string, denoted by Ai. A temporary variable called Ci is used to name the string that represents the outcome number. For the following round, the names of the two components are switched, resulting in the modified Ai, or Ci, becoming Bi + 1 and Ai becoming Bi + 1.

The Feistel structure for encryption and decryption is nearly interchangeable. There are three variations:

I.   The order of the round indices is changed;
II.  The roles of the two components of the data in the round function are switched, so that, in addition to n, T, and i, Ai + 1 (instead of Bi) is used as the input to FK, and Bi + 1 (instead of Ai) is combined with the output to produce Ai (instead of Bi + 1)
III. Modular addition is changed to modular subtraction (the output of FK from Bi +1).

The solution outlined in this article employs the NIST standard format-preserving encryption techniques FF1 and FF3-1.

### 3.2. FFX Scheme Implementation

Figure 2 illustrates the structure of the airborne position message. The downlink format (DF), capability (CA), ICAO address, payload, and parity identity (PI) are the five fields that make up the 112 bits (14 bytes) [26]. The DF field, which has a constant value of 17, describes the message format (10001 in binary). These bits do not undergo encryption, allowing position reports to be recognized in every message. The other fields may be secured. To make it simpler to provide byte-aligned input, the downlink format and capability fields remain unencrypted, since they are constant in airborne mode [12]. Fifty-two (52) bits are equally distributed across the A and B parts of the message and can be used as a Feistel-based encryption function. However, encrypting the ICAO address field is also proposed as a possible solution to conceal information by disconnecting the link between the ICAO address field and other fields [24].

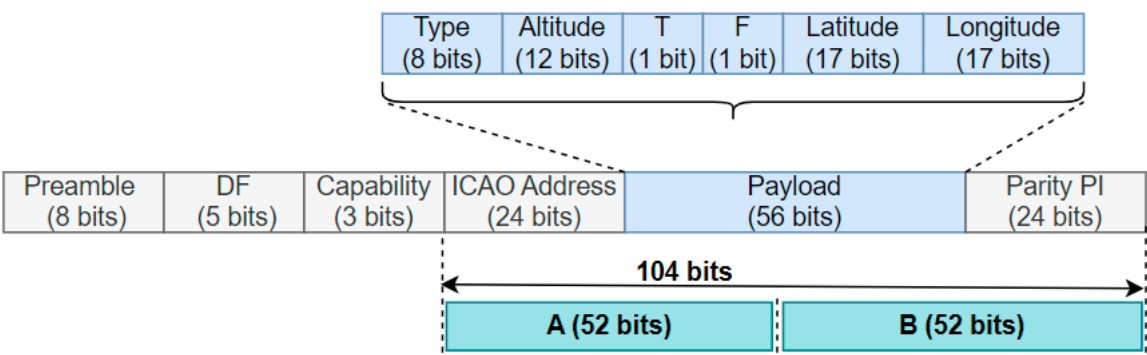

**Figure 2.** ADS-B message structure and split to use as FFX input [26].

Overall, the following test scenarios are going to be explored in this research. Therefore, the following numbers of bytes are encrypted for each test case.

I.   14-Byte (whole message encryption)
II.  13-Byte (excluding DF and CA fields)
III. 8-Byte (ICAO and PI fields)
IV.  3-Byte (ICAO field)

FFX encryption and decryption functions must be added in both the ADS-B In and the ADS-B Out systems. Figure 3 illustrates SADS-B Out subfunction blocks. Positioning and

velocity information are received from the navigation system via the interface function. The message then encodes and assembles. In this stage, the message is encrypted by using the FFX algorithm and a symmetric key. The preamble message then attaches to the encrypted message. In the last stage, the message is scheduled based on the message transmission rate to be broadcast by the transmitter block.

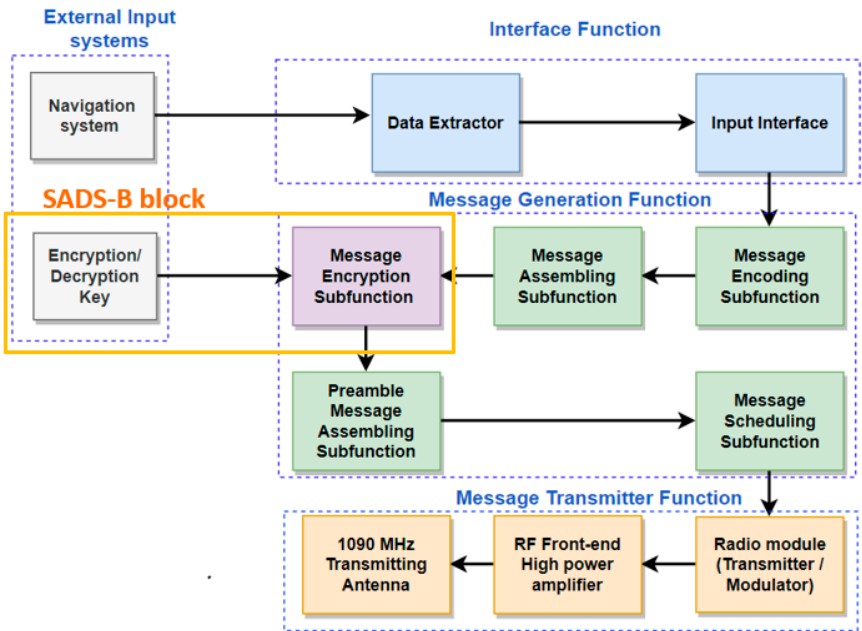

**Figure 3.** SADS-B Out subfunction block diagram.

Similarly, Figure 4 illustrates SADS-B In subfunction blocks. ADS-B Out signals are the input of the message receiver function. After demodulation, preamble pulse detection and enhanced bit detection blocks detect preamble pulse and data bits, respectively. In this phase, the detected burst of data is decrypted by using the FFX algorithm and the symmetric key. It should be noted that the data input of the FFX algorithm is one of the above test scenarios for 14, 13, and 8 bytes. The 112 bits of data can thus be verified as CRC error, and in case of error, the corresponding correction is applied. The message is ultimately decoded and displayed.

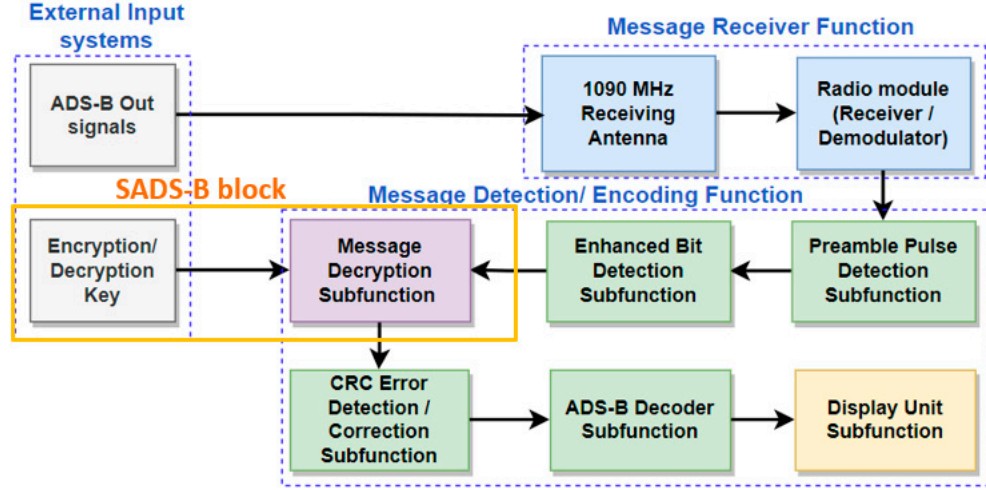

**Figure 4.** SADS-B In's subfunction block diagram.

### 3.3. Blockchain-Based Key Exchange Solution

A blockchain, originally known as block chain, is a growing list of records called blocks. Blockchain consists of a sequence of blocks, and the latter block is linked to the previous one by including the cryptographic hash of the previous block. In addition, the block also contains Index, Time-stamp, Previous Hash, Hash, and Data [27]. All nodes sync using a gossip protocol, which is illustrated in Figure 5. When you change any of these data, you will change the entire block, and the subsequent blocks will notice that something has changed.

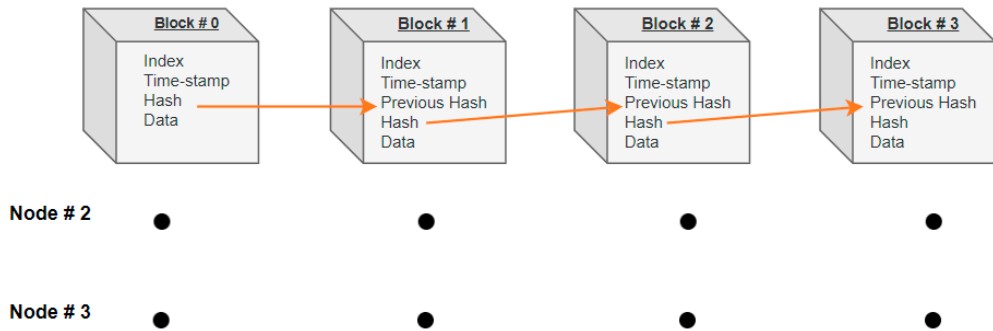

**Figure 5.** Example of Blockchain Blocks.

The four main types of blockchain networks are public, private (or managed), consortium, and hybrid blockchains. The types are selected based on the access, confidentiality, efficiency, and trust requirements. For ADS-B application, it is desirable to have private and permissioned blockchains to satisfy the following requirements:

- Access—trusted members need permission to join;
- Confidentiality—read/write access limitation among members;
- Efficiency—higher efficiency in transactions; and
- Trust—the degree of trust.

Hyperledger Fabric (Fabric) is considered the best option, as the framework offers excellent security and performance as well as adaptable tools for access management, privacy, and integrating business logic. A wide variety of use cases specifically make use of Fabric's characteristics [28,29].

Our goal is to explore how blockchains can protect the sharing of sensitive data. To be more specific, we want to highlight a high-level blockchain-based concept that offers a decentralized nature for flight information, particularly for enabling blockchain technology for key exchange and the positioning of information storage. Figure 6 shows the blockchain network topology for key and data exchange. Node 1 is considered as the administrator and thus can add each node to the network and assign private and public keys to each node for data exchanges. Other nodes use the public key of node 1 to encrypt data based on the RSA scheme and publish it via the MQTT protocol to the admin, Node 1, which receives the information and decrypts the data via an RSA private key, and then the data is stored in the blockchain nodes. In addition, an FFX encryption/decryption key is encrypted by admin node according to each node's corresponding public key, and it is published to the node. Thus, every node receives the key via a secured channel.

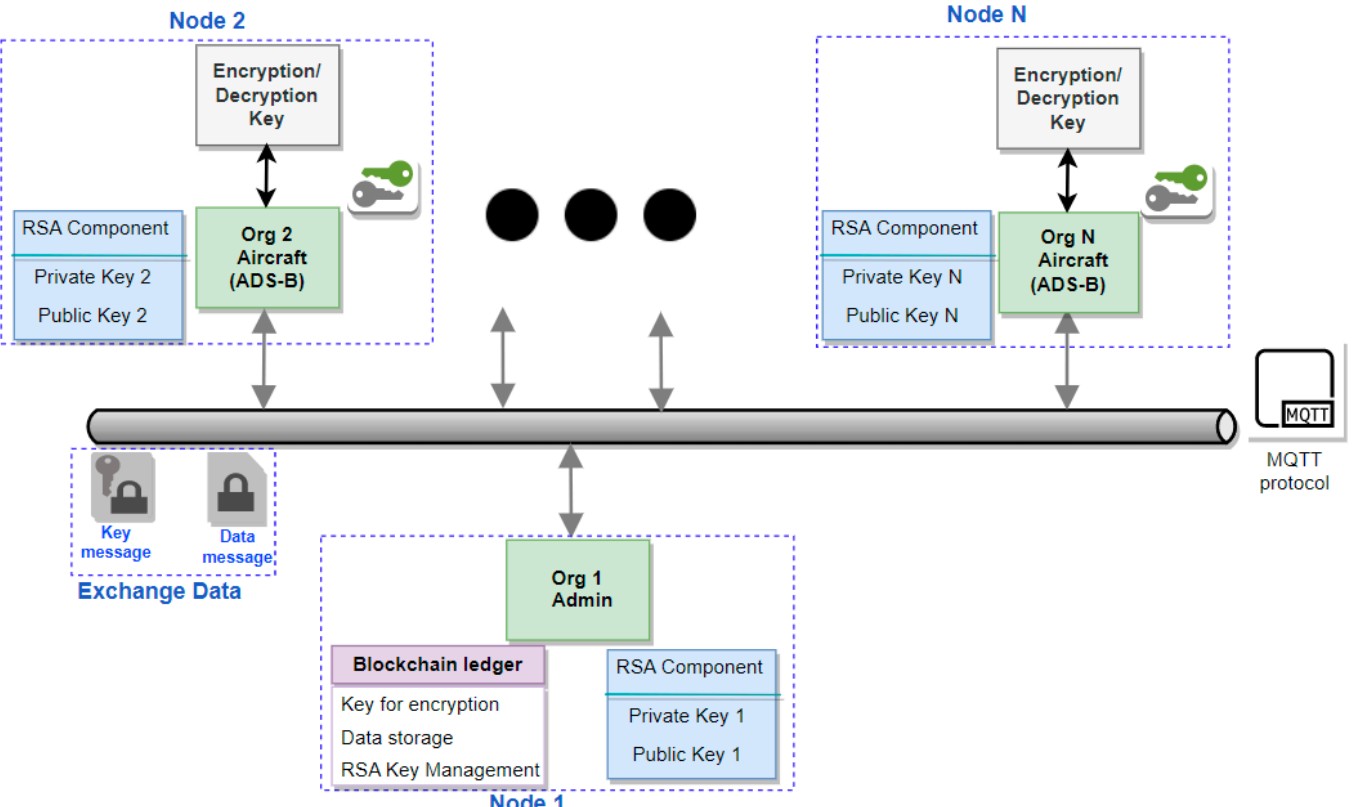

**Figure 6.** Blockchain network topology for key and data exchange.

## 4. Experimental Results

Detailed experiments were performed to evaluate system performance and to measure security levels. These experiments assessed the following four aspects:

- Entropy and security complexity: In cryptography, entropy is frequently used to measure a bit string's level of uncertainty. In addition, serial correlation and plain and encrypted data correlation are key indicators for security measurement.
- Sensitivity: The minimum trigger level (MTL) of an ADS-B receiver processing signals within the frequency span of 1089 to 1091 MHz must respect the MTL constraint, which is −84 dBm for ADS-B class "A3" [26].
- Computational time: Encryption increases the time required for message transmission and reception at both ends. Therefore, it is essential to verify the impact of such increased processing time on both ADS-B Out/In.
- Waveform comparison: Undoubtedly, applying the FFX algorithm alters the bit sequence in time, so it is wise to observe the spectrum accordingly.

Entropy measurements were performed on both plain data and encrypted data of real flight tests according to ADS-B Out logs. This flight test was performed in a light aircraft (Cessna C-172) flying from Saint-Hubert (CYHU) Airport to Mirabel (CYMX) Airport, as shown in Figure 7 [30].

It is worth noting that the entropy is only computed for the "odd" and "even" CPR position messages. In addition, the entropy of messages is measured in bits per byte. The entropy or information content is computed according to Formula (1).

$$H(X) = - \sum_{x_i \in X} p(x_i) log_2 p(x_i) \qquad (1)$$

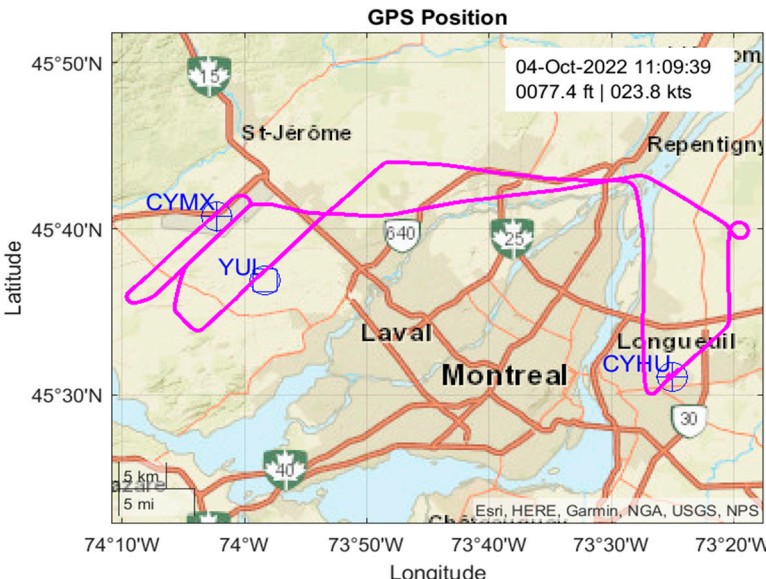

**Figure 7.** Flight test route.

Figure 8 illustrates plain and cipher data entropy for various scenarios. For the 14-byte experiment, the cipher data produced an entropy of 7.9985 bits per byte, whereas the plain text messages had an entropy of 6.7036 bits per byte. These results are similar to the experimental results found by Finke et al. [12]. Moreover, as the cipher data length decreases, the entropy value decreases accordingly. A low entropy value signifies that encrypted data is more vulnerable to possible attacks.

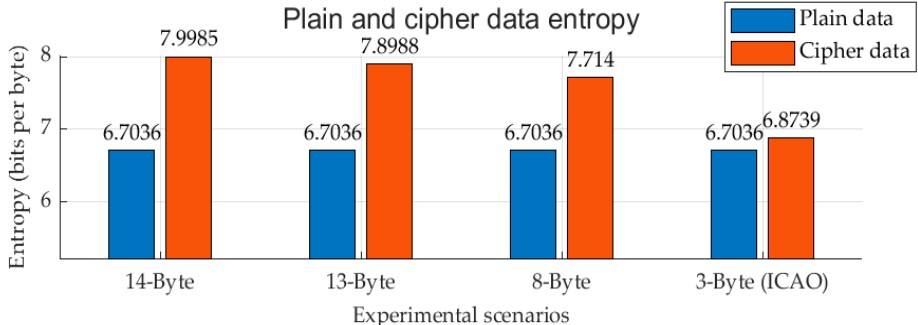

**Figure 8.** Plain and cipher data entropy.

Apart from entropy, the serial correlation of plain and cipher data and the correlation between plain and cipher data are extracted from ADS-B Out flight information, both of which are presented in Figure 9. As revealed in Figure 9, there is a strong serial correlation among the plain data, while the cipher data serial correlation value has an inverse relation with the length of the cipher data. More specifically, the cipher data's serial correlation value with longer data length is less than its correlation with shorter data size. As expected, plain and cipher data correlation follows the same trend as cipher data serial correlation. Therefore, a higher correlation value, particularly in the case of ICAO encryption, is more vulnerable to codebook attacks.

The second experiment was performed to evaluate both standard ADS-B In and SADS-B In signal sensitivity. Figure 10a is a diagram of the test bench designed to evaluate signal sensitivity [30]. As the block diagram illustrates, the ADS-B Out generates ADS-B signals, which are then weakened by a variable attenuator. Next, a splitter provides two signal outputs with the same signal strength, Out1 and Out2. Out1 is used as the input of ADS-B In and Out2 is used as the spectrum analyzer input for signal power measurements. The same test is done for SADS-B. The graph in Figure 10b shows that there are no changes in

either standard ADS-B's or SADS-B's sensitivity criteria. The Accuracy is calculated based on the following formula:

$$Accuracy = 100 * \frac{(count\ of\ correctly\ received\ ADSB\ packets)}{(total\ count\ of\ tranmissted\ ADSB\ packts)} \tag{2}$$

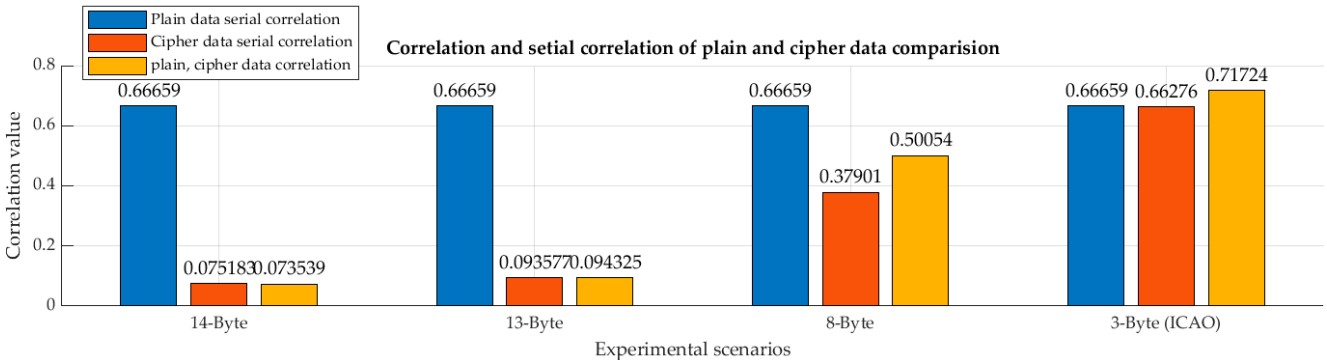

**Figure 9.** Correlation and serial correlation of plain and cipher data.

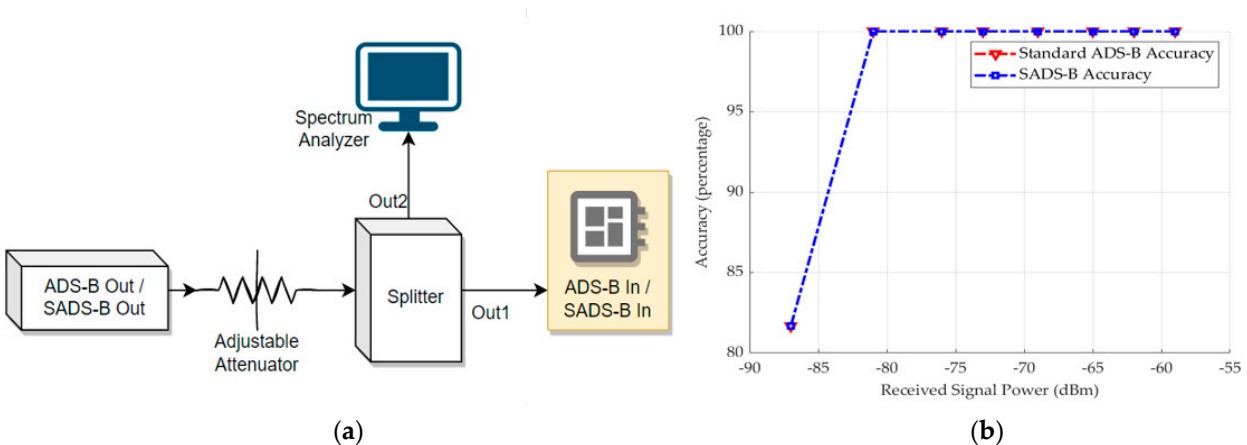

**Figure 10.** (**a**) Sensitivity test bench block diagram, (**b**) Measured accuracy of standard ADS-B and SADS-B signals.

A third experiment was conducted to measure the computation time for both standard ADS-B and SADS-B. Table 2 lists the mean values of the computational time for both systems in MATLAB Simulink and C++. As expected, the SADS-B processing time is higher than the standard time. However, this does not lead to any drawback in SADS-B Out, since the processing time (0.084 milliseconds) is far less than the minimum transmission rate (0.5 s). The SADS-B In performance does not show any degradation during this test, and its processing time is less than 120 microseconds. However, executing SADS-B SDAM along with other SDAMs leads to longer processing times, and its performance might be affected by a high volume of ADS-B traffic. This performance issue must be addressed.

**Table 2.** Mean value of computational time of standard ADS-B and SADS-B in Simulink and C++.

| Software | MATLAB Simulink | | C++ | |
|---|---|---|---|---|
| System type | ADS-B | SADS-B | ADS-B | SADS-B |
| Time (millisecond) | 1.107 | 3.104 | 0.049 | 0.084 |

Spectrum verification was done to compare the time domain and frequency domain signals simultaneously. As Figure 11 illustrates, there is no harmony or strong signal out of the ADS-B band. Thus, SADS-B has not produced interference.

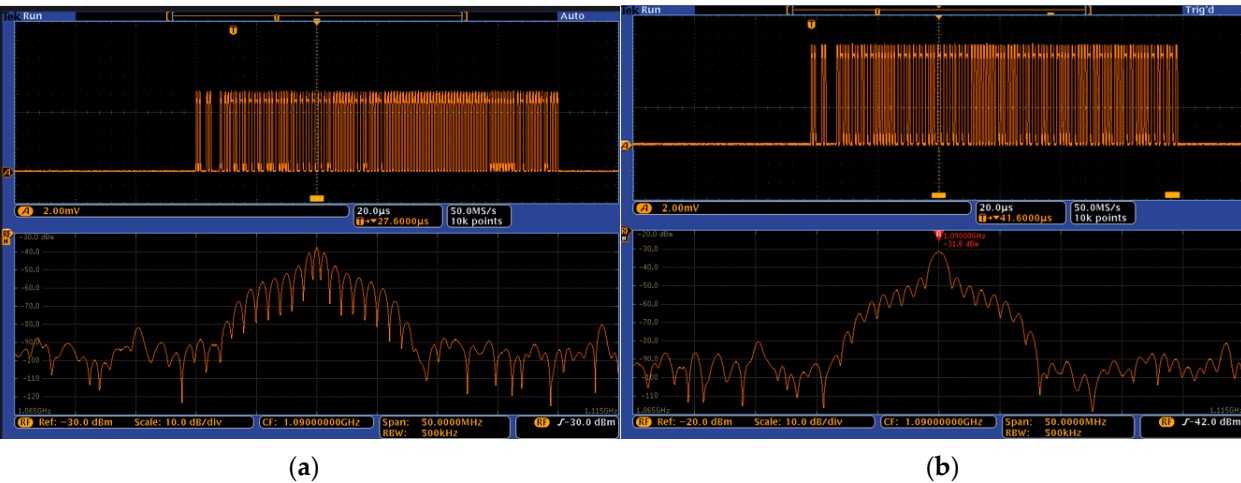

**Figure 11.** (**a**) Standard ADS-B waveform and spectrum, (**b**) SADS-B waveform and spectrum.

To sum up, the result of entropy and serial correlation proves that the proposed algorithm offers a sufficient level for both factors. The higher entropy and the lower serial correlation, the more security of the data. Therefore, the ADS-B message integrity and confidentiality will be guaranteed. The proposed blockchain scheme is not only used for exchanging the key, but is also utilized as a database to store ADS-B Out data. The ADS-B In after receiving the data can verify and validate with the stored database. Therefore, the authentication of each node can be performed indirectly without adding an extra payload to ADS-B messages.

## 5. Discussion

The goal of this research was to address ADS-B threats based on FPE methodology and to design and implement SADS-B by deploying SDR. Blockchain features were utilized to manage key exchanges for encryption purposes within ADS-B nodes. System performance and major key requirements were compared with standard ADS-B.

FFX was used here for encryption purposes, as it is a NIST standard algorithm, particularly FF1 and FF3-1. It has already been used for the exchange of sensitive information, including credit card numbers, ID numbers, and social insurance numbers. In addition, criteria for the measurement of the security complexity of FFX algorithms are defined for ADS-B for security level evaluation. For this purpose, the entropy of ADS-B information is computed for both standard ADS-B and SADS-B for various scenarios. The results indicate that the encryption of longer ADS-B data leads to higher entropy and higher security. In addition, after considering other factors such as serial correlation and plain and cipher data correlation, the optimal approach is to encrypt either 13- or 14-byte data sizes.

Apart from security concerns, a sensitivity experiment was conducted to assess system performance and to compare the results with standard ADS-B. No changes in system sensitivity were observed; therefore, the FFX-integrated system is capable of detecting poor signals such as that of standard ADS-B.

Perhaps the most cited advantages of the FAA's switch to ADS-B surveillance are the quick transmission rate and the related controller scope refresh rate. As mentioned in Figures 3 and 4, FFX adds an extra block of encryption and decryption subfunction to ADS-B In and ADS-B Out, respectively. The encryption increases the time required for message transmission and reception at both ends.

Since the message generation time in SADS-B Out is far less than the transmission rate, adding an encryption block does not affect the SADS-B operation. However, the performance of SADS-B In might be challenged when there are many SADS-B signals. The computational time takes longer in SADS-B because ADS-B is based on cascading blocks.

In other words, each block operation depends on the previous block. This issue could be addressed by applying a threading feature to the FFX algorithm.

Blockchain is deployed to provide a secured channel for key exchanges among the ADS-B nodes. Since Hyperledger fabric satisfies the requirements of ADS-B and has been deployed in known cases, it is selected here for key exchange and for sensitive information storage. The key could be scheduled to update periodically.

Table 3 offers a comparison of the different types of solutions in terms of their implementation possibility (Achievability), security level, sensitivity degradation, and compatibility. Based on this comparison, FPE of size 14 and 13 bytes qualifies as the most achievable, as it is easy to implement, offers a high level of security without any drawback in sensitivity, and is compatible with standard ADS-B and only needs to be added to the software.

**Table 3.** Comparison of various solutions.

| Category | Achievability | Security Level | Sensitivity Degradation | Compatibility |
|---|---|---|---|---|
| Public key | Hard | High | Yes | Requires key management, agreement modification |
| Message Authentication Code | Easy | Low | Yes | Requires key management, agreement modification |
| SADS-B with PSK | Medium | Low | Yes | Requires key management, additional software |
| FPE of size 14-, 13-byte | Easy | High | No | Requires key management, additional software |
| FPE ICAO only | Easy | Low | No | Requires key management, additional software |

Table 4 represents a comparison of the different types of solutions in terms of their security requirements, including data integrity, location integrity, confidentiality, and authentication. Based on the comparison, the paper solution qualifies to achieve all requirements without adding any extra payload to ADS-B data.

**Table 4.** Security requirement comparison of various solutions.

| Category | Data Integrity | Location Integrity | Confidentiality | Authentication | Remarks |
|---|---|---|---|---|---|
| Public key | Y | Y | Y | Y | Adding extra load |
| Message Authentication Code | N | N | N | Y | Adding extra load |
| SADS-B with PSK | N | N | N | Y | |
| Holistic framework | Y | Y | Y | Y | Adding extra load |
| SVM | P (detect injection) | N | N | N | P = partially addressed |
| Physical layer protection | P (detect injection) | Y | N | N | P = partially addressed |
| supervised deep learning | P (detect injection) | Y | N | N | P = partially addressed |
| FPE & Blockchain (this paper) | Y | Y | Y | Y | |

On the one hand, implementation of the FFX algorithm secured the confidentiality and the integrity of ADS-B messages. Therefore, only the authorized nodes are capable of decoding the encrypted ADS-B data. The transmitted data is encrypted by a standard methodology, and it has high entropy to be decoded, thus attackers are not able to decode the data and manipulate it. The solution guarantees the integrity of the ADS-B data. On the other hand, the proposed permissioned blockchain model is used to share the encryption key among nodes (ADS-B systems), and the frequency of the key updating will secure the system confidentiality and integrity. In addition, the blockchain system provides a data repository in order to update information for each ADS-B Out data, and the receiver can verify the received data from the air with blockchain data and perform authentication. Therefore, the proposed model addresses the confidentiality, integrity, and authenticity concerns.

## 6. Conclusions

Deploying secure ADS-B leads to the improved protection of data and information against potential threats to ADS-B, including eavesdropping, message injection, and deletion. This paper assesses the FFX algorithm's features and the possibility of its implementation in both Simulink and C++ with SDR BladeRF for both ADS-B Out/In systems. Real flight data entropy, serial correlation, and plain and cipher data correlation were selected as metrics with which to evaluate the system security level for various scenarios. The FFX

algorithm provides sufficient diffusion and confusion to protect sensitive information in ADS-B data. Moreover, the ADS-B system capacity is not reduced, as message data length is preserved. In addition, the effects of SADS-B on standard ADS-B performance were explored. Although the SADS-B sensitivity performance was not altered, its computational time was increased. As explained in the discussion section, SADS-B Out's performance is not impacted by this computational time increment. The SADSB In computational time could be enhanced by applying a threading feature on the FFX algorithm to alleviate processing time issues due to cascading blocks.

The novelty of this work is summarized as follows:

- Investigating the FFX algorithms' impact on ADS-B messages and how its utilization brings about security advantages to ADS-B sensitive data.
- Designing and implementing FFX algorithms in an SDR system (BladeRF) which is compatible with the standard ADS-B for both the transmitter and receiver.
- Offering a new solution for the key exchange for encryption via the blockchain scheme
- Experimenting with lab and flight tests without any degradation in standard ADS-B performance and key requirements.

It might be argued that, by applying FPE encryption, ADS-B's desirable openness feature will be lost, and this is mostly true. However, aviation authorities can implement the FFX algorithm as a contingency response during active attack circumstances. This would protect ADS-B systems from any serious consequences due to attacks.

Although the work examined SADS-B Out/In in an operational environment, performing experimental tests on a large scale with a high number of various aircraft to validate its performance is required to convince the aviation industry to utilize the advantages of SADS-B. In addition, it is suggested to establish the experimental procedure for examining the proposed system behaviour against the various threats and attacks.

**Author Contributions:** J.H.M. was responsible for the algorithm design, prepared the literature review, and wrote the contents of the paper. R.J.L. is the lead project scientist with diverse contributions from the algorithm to overall architecture while supervising the project. The experiments, simulations, analysis, and conclusion were performed by J.H.M. Writing—review & editing, A.A. Project administration, J.-M.G. All authors have read and agreed to the published version of the manuscript.

**Funding:** This research was funded by industry partners.

**Institutional Review Board Statement:** Not applicable.

**Informed Consent Statement:** Informed consent was obtained from all subjects involved in the study.

**Acknowledgments:** I would like to sincerely thank my supervisor, Rene Jr Landry, who provided all of the necessary resources and equipment to complete this research.

**Conflicts of Interest:** The authors declare that they have no conflict of interest.

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
