# Peer review of "Security Establishment in ADS-B by Format-Preserving Encryption and Blockchain Schemes"

_applsci, doi:10.3390/app13053105_

Round 1

Reviewer 1 Report

What is the actual finding of the research, include in the abstract section. Abbreviation of  FFX is not discussed properly. The author should include the proper abbreviations at the first appearance.

What are the state of the art models available in existing work, those models are not discussed in the introduction section. Provide citation.

Problem statement is not concluded properly, the author should consolidate the problem statement and research gap of literature.

The FFX a common algorithm? What is the novelty here? 

How the problems are solved, the discussion is missing.

The author mentioned the attacks in the introduction and related works but the analysis related to attacks is missing in the result section. Include it.

Recent year papers are missing, the author should include 2022 papers to support the research.

The author should improve the comparative analysis by adding a few more papers.

Author Response

Dear All,

Your precious time and valuable feedback are highly appreciated. Kindly find the responses in blue color in the attached file.

Thanks again for your time and valuable remarks.  

Reviewer 2 Report

The present paper highlight the limitations of legacy ADS-B systems address the possibility of developing format-preserving encryption.

 There are several weaknesses, as follows.

 - The state of the art is based on 26 reference titles, out of which only 5 are very new. Taking into account the high dynamics of the domain, I’d suggest the authors to search newer papers published on similar thematic and update their references with newer titles, published in important Journals in the last 2-3 years.

 - regarding The Feistel structure (section 3.1 and figure 1) – references are needed.

 - same for structure of the airborne position message. (section 3.2 and figures 2,3,4)  

 - a table with abbreviations should increase the clarity of the text.

 - in section 4 – can the authors present a formula for the entropy (Figure 8)?

 -  same for accuracy (Figure 10b)

 - In Figure 11 the text cannot be seen clearly o the figures.

 - the results has to be explained more in depth (why those results, what are they showing, what is new in this approach)

 - In the conclusion section the authors should emphasize more the novelty of the present approach. They should emphasize what is beyond of the existing state of the art in this paper and how the results can be further developed, because at this point this is not very clear.

 - Also, a comparison with other similar ones from existing literature is necessary.

Author Response

(The authors gave the same response as above.)

Round 2

Reviewer 2 Report

After answering the reviewers questions and completing the manuscript the paper improved, and I consider that now it can be published.